# FAKE SENTENCE DETECTION AS A TRAINING TASK FOR SENTENCE ENCODING

## ABSTRACT

Sentence encoders are typically trained on generative language modeling tasks with large unlabeled datasets. While these encoders achieve strong results on many sentence-level tasks, they are difficult to train with long training cycles. We introduce fake sentence detection as a new discriminative training task for learning sentence encoders. We automatically generate fake sentences by corrupting original sentences from a source collection and train the encoders to produce representations that are effective at detecting fake sentences. This binary classification task turns to be quite efficient for training sentence encoders. We compare a basic BiLSTM encoder trained on this task with strong sentence encoding models (Skip-thought and FastSent) trained on a language modeling task. We find that the BiLSTM trains much faster on fake sentence detection (20 hours instead of weeks) using smaller amounts of data (1M instead of 64M sentences). Further analysis shows the learned representations also capture many syntactic and semantic properties expected from good sentence representations.

## 1 INTRODUCTION

Universal sentence encoding is a way to scale language processing tasks, especially when the target training data is limited. Solutions to sentence-level language processing tasks can be seen as consisting of two parts – one that creates a generic representation which approximates the meaning of the sentence, and another that uses this representations to make the target classifications. The idea behind universal sentence encoding is to learn generic sentence representations from unlabeled texts, which are often much larger and easier to obtain than the training data for the target tasks. A good universal encoding is one which is effective for training downstream target tasks.

The success of language modeling ideas for learning word representations has inspired similar ideas for learning universal sentence representations. Skip-thought model (Kiros et al., 2015) uses a Bidirectional LSTM (BiLSTM) encoder, which encodes a given sentence, and uses the encoding to generate neighboring sentences. Following up on this general idea, the FastSent model (Hill et al., 2016) simplifies the task and the encoder further by learning to output the average word embeddings so that they can predict words in the neighboring sentences (but not necessarily in a sequence).

This language modeling based training is undesirable in two respects. (1) Predicting neighboring sentences is a difficult *generative training* task. The task requires having a large output space, at least the size of the vocabulary. It also requires a complex decoding model such as an LSTM decoder with millions of parameters. Effective training requires a large amount of training data with long training cycles. Indeed Skip-thought requires tens of millions of training sentences and it takes more than a week to train. (2) Even minor changes to a sentence can drastically change its meaning, a phenomenon that needs to be modeled for many NLP tasks. Language model based training essentially relies on the training sample for this kind of generalization, i.e., the training text collection should include many instances of sentences that have only minor lexical differences but found in completely different contexts. Larger the dataset the more likely it is for the encoder to learn how to model these fine grained distinctions.

In this work, we introduce a *discriminative training* task *fake sentence detection* to address these challenges. The main idea is to generate fake sentences by corrupting an original sentence. We use two methods to generate fake sentences: *word shuffling* where we swap the positions of two words at random and *word dropping*, where we drop a word at random from the original sentence.

The resulting fake sentences are mostly similar to the original sentences—a fake sentence differs from its source in at most two word positions. We create the training corpus from a source corpus of unlabeled sentences. For each sentence in the source corpus we add multiple fake sentences by corrupting the source sentence.

This training task has three key advantages: (1) This binary classification task can be modeled with fewer parameters in the output layer and can be trained more efficiently compared to language modeling based training tasks. (2) From a language standpoint, the task forces the encoder to track both syntax and semantics. For instance, swapping words within a sentence can break the syntax (e.g., "John landed in Chicago on Friday" versus "John landed Chicago in on Friday"), and break or alter the semantics; it can lead to an incoherent or less plausible sentence (e.g., "John landed in Chicago on Friday" versus "Chicago landed in John on Friday"). (3) The task explicitly forces the encoder to capture big shifts in meaning that arise from small changes to a sentence. In particular, since the discrimination is really dependent on a small change to an original sentence, encoders cannot get away with simple aggregation—they need to model compositional aspects well enough to be able to detect small but semantically shifts in sentence constructions.

In overview, we train a bidirectional LSTM (BiLSTM) as our universal encoder and use a three-layer feed-forward network that uses the encoded sentence to predict if it is fake or real. We then evaluate this trained encoder *without any further tuning* on multiple sentence-level tasks and use probing tasks (Conneau et al., 2018) to study the syntactic and semantic properties in the encoded representations.

In summary, this paper makes the following main contributions:

1. Introduces fake sentence detection as an unsupervised training task for learning sentence encoders that can distinguish between small changes in mostly similar sentences.

2. Provides an empirical evaluation on multiple sentence-level tasks showing representations trained on the fake sentence tasks outperform a strong baseline model trained on language modeling tasks, even when training on small amounts of data (1M vs. 64M sentences) reducing training time from weeks to within 20 hours.

3. Demonstrates that the fake sentence training produces encoded representations that are better at capturing many linguistic properties compared to language model training.

## 2 MOTIVATION

Many sentence-level prediction tasks require learning a function to estimate the probability of a label $y$ given a sentence $x$. This function can be parameterized as $P(y|x, \theta)$, and the parameter vector $\theta$ can be learned by minimizing the negative conditional log likelihood of the training data $\mathcal{D}$, i.e., $\min_\theta - \sum_{(x,y)\in\mathcal{D}} \log(P(y|x,\theta))$. However, the amount of training data is limited in many cases, so learning $\theta$ by optimizing the conditional probability leads to overfitting, especially if one uses a deep neural network with millions of parameters.

One approach to address this problem is to break the task into two sub-tasks each with its own set of parameters: (1) sentence encoding—produce an effective low-dimensional representation of the sentence $enc(x, \theta_1)$, (2) target prediction—estimate probability of the label $y$ conditioned on the encoding $P(y|x, enc(x, \theta_1), \theta_2)$. The dimension of $\theta_2$ is much smaller than the dimension of $\theta$ and it can be learned with the provided labeled training data $\mathcal{D}$. For the encoder function $enc(x, \theta_1)$, it is hoped that the parameter vector $\theta_1$ can be learned or pre-trained with a different sentence-level task using other data, which is possibly unlabeled, but can be easily collected in large quantity. The encoder can be trained on the auxiliary task as part of a prediction function $f$ which has its own set of parameters $\theta_3$. The encoder parameters $\theta_1$ and the auxiliary task parameters $\theta_3$ are learned to minimize the loss on the auxiliary task: $\sum_{x\in\mathcal{U}} \mathcal{L}_{aux}(f(enc(x, \theta_1), \theta_3))$.

In this work, we are interested in learning an universal sentence encoder that can be subsequently used for various downstream sentence-level tasks. One reasonable choice for the auxiliary loss function would be the negative log likelihood of the data, i.e., $-\log P(x|\theta_1, \theta_3)$ or equivalently $-\log(P(enc(x, \theta_1)|\theta_1, \theta_3))$. Recall for the downstream task, we will seek the parameter vector $\theta_2$ to minimize the negative conditional log likelihood $-\log(P(y|enc(x, \theta_1), \theta_1, \theta_2))$. The combination of two loss functions is: $-\log(P(y|enc(x, \theta_1), \theta_1, \theta_2)) - \log(P(enc(x, \theta_1)|\theta_1, \theta_3))$. This is

equivalent to $-\log(P(y, enc(x, \theta_1)|\theta_1, \theta_2, \theta_3))$. In other words, we are maximizing the joint probability of observing the label $y$ and the encoding vector $enc(x, \theta_1)$. Optimizing the joint probability of the label and data corresponds to having a generative classifier, as opposed to the discriminative classifier that optimizes a conditional probability. As shown by (Ng & Jordan, 2002), a generative classifier converges faster to its asymptotic behavior than a discriminative classifier does. A generative approach is particularly useful when there is limited amount of training data. Thus, we propose to learn a sentence encoder by maximizing the data likelihood $P(enc(x, \theta_1)|\theta_1, \theta_3)$, and we expect it to be an useful universal sentence encoder, especially when there is limited amount of labeled training data for downstream sentence-level task.

It is surprisingly simple to learn an encoder to maximize the data likelihood $P(enc(x, \theta_1)|\theta_1, \theta_3)$, and this paper proposes exactly that. We can parameterize this probability using a simple function such as a log-linear model:

$$P(enc(x, \theta_1)|\theta_1, \theta_3) = \frac{1}{Z}\exp(-\theta_3^T enc(x, \theta_1)) \tag{1}$$

where the functional form of the encoder $enc$ is an BiLSTM, and $Z$ is a normalization constant to ensure a valid probability function.

$$Z = \sum_x \exp(-\theta_3^T enc(x, \theta_1)) = \sum_{x \in \mathcal{U}} \exp(-\theta_3^T enc(x, \theta_1)) + \sum_{x \in \mathcal{V}} \exp(-\theta_3^T enc(x, \theta_1)). \tag{2}$$

where $\mathcal{U}$ is the set of real sentences, and $\mathcal{V}$ is the set of fake sentences. Our objective is to learn $\theta_1, \theta_3$ to maximize the quantity in Equation 1. This is equivalent to maximizing $\sum_{x \in \mathcal{U}} \exp(-\theta_3^T enc(x, \theta_1))$ while minimizing $\sum_{x \in \mathcal{V}} \exp(-\theta_3^T enc(x, \theta_1))$. This can simply be done by training a classifier to separate real sentences from fake sentences.

In this work, we approximate the space of fake sentences by randomly swapping or dropping words from real sentences. Our method has some connection to the GAN framework (Goodfellow et al., 2014), where there is a generator that attempts to generate realistic data and a discriminator that distinguishes between real and generated data instances. As shown by Goodfellow et al. (2014), this approach will converge to a solution where the output of the discriminator is the estimated value for a data point to be real. In our case, we do not need to learn the generator and the discriminator in an adversarial manner because we already have a 'strong' generator that generates highly realistic sentences, simply by shuffling or dropping words. Furthermore, our method can be thought as a generative model, defined and trained using a discriminative function. This aspect is related to the recently proposed Introspective Neural Networks (Lazarow et al., 2017).

There are some similarities between the proposed method and Skip-thought sentence encoder. The training task of Skip-thought involves predicting the next or previous sentence $z$ that follows or precedes the current encoded sentence $x$. This sentence language model is realized using an LSTM decoder, which sequentially generates a word at each position $i$ conditioning on the previous encoded sentence $enc(x)$, and the words that have been decoded so far $(z_1, \cdots, z_{i-1})$. This model can be seen as maximizing the conditional probability:

$$P(z|enc(x, \theta_1), \theta_3) = \prod_{i=1}^{|z|-1} P(z_i|z_{1:i-1}, enc(x, \theta_1), \theta_3). \tag{3}$$

Interestingly, both Skip-thought and our proposed method model the unlabeled data $\mathcal{U}$, a sequence of sentences $u^1, u^2, \cdots, u^m$, and aim to optimize for the joint probability distribution: $P(\mathcal{U}|\theta_1, \theta_3)$. In the case of Skip-thought, there is an underlying Markovian assumption that the probability of a sentence only depends on the previous sentence. That is to approximate $P(\mathcal{U}|\theta_1, \theta_3)$ by $\prod_k P(u^k|enc(u^{k-1}, \theta_1), \theta_3)$. In the case of the proposed method, we assume the sentences are independent of one another, so $P(\mathcal{U}|\theta_1, \theta_3)$ can be approximated as $\prod_k P(enc(u^k, \theta_1)|\theta_1, \theta_3)$. Both methods are generative models of the data, and both are useful sentence encoders for the downstream tasks, as will be seen in the experiment sections.

However, Skip-thought model has several disadvantages compared to the proposed method: (1) The architecture of Skip-thought model is much more complex. Skip-thought model requires an LSTM decoder with millions of parameters. Furthermore, the need for computing the probability of individual words requires having a large output space, as large as the size of the vocabulary.

This further increases then number of parameters and the complexity of the overall architecture. (2) The modeling task of Skip-thought is much harder than that of the proposed method. Skip-thought requires a probability model for both complete and incomplete sentences, and this much harder than modeling the probability of complete sentences alone (as proposed here). (3) Due to the complexity of the architecture and the difficulty of the modeling task, Skip-thought requires much more training data and longer training time. (4) Skip-thought is a purely generative model and it is only trained using the real data. Meanwhile, the proposed method is a generative model defined via a discriminative function. This function can be trained using both real and fake data, where the fake data can be selectively chosen to be very hard examples. Specifically, we use word shuffling and word dropping to generate hard examples that do not differ from the original sentences. This enables the sentence encoder to focus on the subtle semantics information.

## 3 RELATED WORK

Previous sentence encoding approaches can be broadly classified as supervised (Conneau et al., 2017; Cer et al., 2018; Marcheggiani & Titov, 2017; Wieting et al., 2015), unsupervised (Kiros et al., 2015; Hill et al., 2016) or semi-supervised approaches (Peters et al., 2018; Dai & Le, 2015; Socher et al., 2011; Clark et al., 2018).

The unsupervised approaches extend the skip-gram (Mikolov et al., 2013) to the sentence level, and use the sentence embedding to predict the adjacent sentences. Skip-thought (Kiros et al., 2015) uses a BiLSTM encoder to obtain a fixed length embedding for a sentence, and uses a BiLSTM decoder to predict adjacent sentences. Training Skip-thought model is expensive, and even one epoch of training on the Toronto BookCorpus (Zhu et al., 2015) dataset takes more than two weeks (Hill et al., 2016) on a single GPU. FastSent (Hill et al., 2016) uses embeddings of a sentence to predict words from the adjacent sentences. A sentence is represented by simply summing up the word representation of all the words in the sentence. FastSent requires less training time than Skip-thought, but FastSent has lower performance on most downstream tasks.

The supervised approaches train the encoders on tasks such as NLI and use transfer learning to adapt the learned encoders to different downstream tasks (Conneau et al., 2017).

Some semi-supervised approaches (Peters et al., 2018; Dai & Le, 2015; Socher et al., 2011) train sentence encoders on large unlabeled datasets, and do a task specific adaptation using labeled data, while some other approaches such as Cross-View Training(CVT) (Clark et al., 2018) jointly train the sentence encoder on the labeled and unlabeled data. CVT applies regular supervised learning on the labeled data, and for the unlabeled data, it enforces consistency across the outputs obtained for partial versions of the input sentence.

In this work, we propose an unsupervised sentence encoder that takes around 20 hours to train on a single GPU, and outperforms Skip-thought and FastSent encoders on multiple downstream tasks. Unlike the previous unsupervised approaches, we use the binary task of real versus fake sentence classification to train a BiLSTM based sentence encoder.

## 4 TRAINING TASKS FOR ENCODERS

We propose a discriminative task for training sentence encoders. The key bottleneck in training sentence encoders is the need for large amounts of labeled data. Prior work use language modeling as a purely generative training task that models the generation of unlabeled text data. The encoder is trained to produce sentence representations which are effective at either generating neighboring sentences (e.g., Skip-thought (Kiros et al., 2015) or to predict the words in the neighboring sentences (Hill et al., 2016). As we outlined in Section 2 this purely generative training raises multiple challenges.

Instead, we propose *fake sentence* detection, a task that targets the same overall generative model, one that generates unlabeled text, but is trained discriminatively. The task requires making a single prediction over an input sentence. In particular, we propose to learn a sentence encoder by training a sequential model to solve the binary classification task of detecting whether a given input sentence is fake or real. This real-fake sentence classification task would perhaps be trivial if the fake sentences look very different from the real sentences. We propose two simple methods to generate noisy

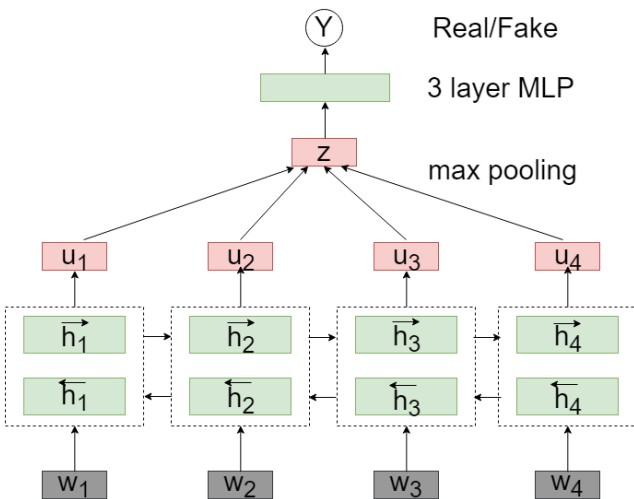

Figure 1: Figure shows the block diagram of the encoder and fully connected layers. Encoder consists of a bidirectional LSTM followed by a max pooling layer. For classification, we use a MLP with two hidden layers.

sentences which look *mostly* similar to real sentences. We describe the noisy sentence generation strategies in Section 4.1. Thus, we create a labeled dataset of real and fake sentences, and train a sequential model to distinguish between real and fake sentences, which results in a model whose classification layer has far fewer parameters than previous language model based encoders.

Our model architecture is described in Section 4.2.

## 4.1 Fake Sentence Generation

For a sentence $X = w_1, w_2, \ldots, w_n$ comprising of $n$ words, we consider two strategies to generate a noisy version of the sentence: **1) WordShuffle**: randomly sample two indices $i$ and $j$ corresponding to words $w_i$ and $w_j$ in $X$, and shuffle the words to obtain the noisy sentence $\hat{X}$. Noisy sentence $\hat{X}$ would be of the same length as the original sentence $X$. **2) WordDrop**: randomly pick one index $i$ corresponding to word $w_i$ and drop the word from the sentence to obtain $\hat{X}$. Note there can be many variants for this strategy but here we experiment with this basic choice.

## 4.2 Real Versus Fake Sentence Classification

Figure 1 shows the proposed architecture of our fake sentence classifier with an encoder and a Multi-layer Perceptron(MLP) with 2 hidden layers. We do not use any nonlinearity after these hidden layers, hence, these layers correspond to the single linear layer represented by $\theta_3$ in Equation 1. Another motivation behind not using the nonlinearity is to keep the classifier simple enought so that the encoder is forced to learn useful representation while training for the fake sentence classification task. The encoder consists of a bidirectional LSTM followed by a max pooling layer. Given a sentence $X = w_1, w_2, \ldots, w_n$ consisting of $n$ words, the forward LSTM of the encoder generates a hidden state vector $\overrightarrow{h_t}$. Similarly, the backward LSTM generates the hidden state $\overleftarrow{h_t}$. The forward and backward hidden states are concatenated to get $u_t = (\overrightarrow{h_t}, \overleftarrow{h_t})$. Processing the entire sentence results in $n$ such vectors, and we apply max-pooling to these concatenated hidden states to obtain vector $z$. $z$ serves as a fixed length encoding of the sentence $X$, which we then use as input to a MLP for classifying the sentence into real/fake classes.

## 5 Evaluation Setup

**Downstream Tasks:** We compare the sentence encoders trained on a large collection (BookCorpus (Zhu et al., 2015)) by testing them on multiple sentence level classification tasks (MR, CR, SUBJ, MPQA, TREC, SST) and one NLI task defined over sentence-pairs (SICK). We also eval-

| Name | Size | Task | No. of Classes |
|------|------|------|----------------|
| MR | 11K | Sentiment | 2 |
| CR | 4K | Product Review | 2 |
| TREC | 11K | Question type | 6 |
| SST | 70K | Sentiment | 2 |
| MPQA | 11K | Opinion Polarity | 2 |
| SUBJ | 10K | Subjectivity | 2 |
| SICK | 10K | NLI | 3 |
| COCO | 123K | Retrieval | - |

Table 1: Downstream tasks and datasets.

| Model | MR | CR | TREC | SST | MPQA | SUBJ | SICK | COCO-Cap | COCO-Img |
|-------|-----|-----|------|-----|------|------|------|----------|----------|
| NB-SVM | 79.4 | 81.8 | - | - | 86.3 | 93.2 | - | - | - |
| FastSent | 70.8 | 78.4 | 80.6 | - | 80.6 | 88.7 | - | - | - |
| Skip-thought (full) | 76.5 | 80.1 | 92.2 | 82.0 | 87.1 | 93.6 | **82.3** | 72.2 | 66.2 |
| Skip-thought (1M) | 65.2 | 70.9 | 79.2 | 66.9 | 81.6 | 86.1 | 75.6 | 51.9 | 46.7 |
| Skip-thought (full) + NB | **80.4** | 81.3 | - | - | 87.5 | 93.6 | - | - | - |
| MC-QT | **80.4** | **85.2** | **92.8** | - | 89.4 | **93.9** | - | - | - |
| WordDrop | 78.8 | 82.2 | 86.6 | **82.9** | **89.8** | 92.7 | 83.2 | 73.8 | 67.3 |
| WordShuffle | 79.8 | 82.4 | 88.4 | 82.4 | **89.8** | 92.6 | 82.3 | 74.2 | 67.3 |

Table 2: Results on downstream tasks: Bold face indicates best result and underlined results show when fake sentence training is better than Skip-thought (full). COCO-Cap and COCO-Img are caption and image retrieval tasks on COCO. We report Recall@5 for the COCO retrieval tasks.

uate the sentence representations for image and caption retrieval tasks on the COCO dataset (Lin et al., 2014). We use the same evaluation protocol and dataset split as (Karpathy & Fei-Fei, 2015; Conneau et al., 2017). Table 1 lists the classification tasks and the datasets. We also compare the sentence representations for how well they capture important syntactic and semantic properties using probing classification tasks (Conneau et al., 2018). For all downstream and probing tasks, we use the encoders to obtain representation for all the sentences, and train logistic regression classifiers on the training split. We tune the $L_2$-norm regularizer using the validation split, and report the results on the test split.

**Training Corpus:** The FastSent and Skip-thought encoders are trained on the full Toronto Book-Corpus of 64M sentences (Zhu et al., 2015). Our models, however, train on a much smaller subset of *only* 1M sentences.

| Model | SentLen | WC | TreeDepth | TopConst | BShift | Tense | SubjNum | ObjNum | SOMO | CoordInv |
|-------|---------|-----|-----------|----------|--------|-------|---------|--------|------|----------|
| Skip-thought (full) | 85.4 | 79.6 | 41.1 | **82.5** | 69.6 | **90.4** | 85.6 | **83.6** | 53.9 | 69.1 |
| Skip-thought (1M) | 54.7 | 33.9 | 30.0 | 60.7 | 58.9 | 85.3 | 76.4 | 70.9 | 51.9 | 61.4 |
| WordDrop | **86.7** | 90.1 | 48.0 | 81.9 | 73.2 | 87.7 | **87.3** | 82.7 | 59.2 | 70.6 |
| WordShuffle | 84.9 | **91.2** | **48.8** | 82.3 | **79.9** | 88.2 | 86.7 | 83.3 | **59.8** | **70.7** |

Table 3: Probing task accuracies. Tasks: SentLen: predict sentence length, WC: is word in sentence, TreeDepth: depth of syntactic tree, TopConst: predict top-level constituent, BShift: is bigram in flipped in sentence, Tense: predict tense of word, Subj(Obj)Num: singular or plural subject, SOMO: semantic odd man out, CoordInv: is co-ordination is inverted.

**Sentence Encoder Implementation:** Our sentence encoder architecture is the same as the BiLSTM-max model (Conneau et al., 2017). We represent words using 300-d pretrained Glove embeddings (Pennington et al., 2014). We use a single layer BiLSTM model, with 2048-d hidden states. The MLP classifier we use for fake sentence detection has two hidden layers with 1024 and 512 neurons. We train separate models for word drop and word shuffle. The models are trained for 15 epochs with a batch size of 64 using SGD algorithm, when training converges with a validation

| Original | Neighbor | ST Rank | FS Rank |
|---|---|---|---|
| I love soccer. | Soccer love I. | 2 | 6 |
| He took her hand and kissed it. | He kissed her hand and took it. | 1 | 4 |

Table 4: Example illustrating how Skip-thought (ST) and Fake sentence (FS) training allows representations to capture big shifts in meaning even with small changes to the sentence form. The table shows where both models rank a specific fake version of a real sentence. The ranking is is obtained using cosine similarity over a random sample of 10000 real and fake sentences when projected via t-SNE.

set accuracy of 89 for word shuffle. The entire training completes in less than 20 hours on a single GPU machine.

**Baseline Approaches:** We compare our results with previous unsupervised sentences encoders, Skip-thought (Kiros et al., 2015), FastSent (Hill et al., 2016), Quick-thought(MC-QT) (Logeswaran & Lee, 2018), and bag-of-words baseline NB-SVM (Wang & Manning, 2012). We use the FastSent and Skip-thought results trained on the full BookCorpus as mentioned in (Conneau et al., 2017). We also report Skip-thought results when trained on a smaller 1M sentence subset.

## 6 RESULTS

**Classification and NLI:** Results are shown in Table 2. Both fake sentence training tasks yield better performance on five out of the seven language tasks when compared to Skip-thought (full), i.e., even when it is trained on the full BookCorpus. Word drop and word shuffle performances are mostly comparable. Skip-thought (1M) row shows that training on a sentence-level language modeling task can fare substantially worse when trained on a smaller subset of data. FastSent, while easier to train and has faster training cycles, is better than Skip-thought (1M) but is worse than the full Skip-thought model.

Further gain in performance can be obtained by validating the performance of the encoder on downstream tasks after each epoch of training, and picking the encoder with best performance on the validation set for the downstream tasks. For MR and SUBJ tasks, this leads to accuracy of 80.3 and 93.3 respectively. For the remaining tasks, performance gain is less significant.

**Image-Caption Retrieval:** On both caption and image retrieval tasks (last 2 columns of Table 2), fake sentence training with word dropping and word shuffle are better than the published Skip-thought results.

**Linguistic Properties:** Table 3 compares sentence encoders for various linguistic properties using the recently proposed probing tasks (Conneau et al., 2018). The goal of each task is to use the input sentence encoding to predict a particular syntactic or semantic property of the original sentence it encodes (e.g., predict if the sentence contains a specific word). Encodings from fake sentence training score higher in six out of the ten tasks.

A key property of the discriminative training is that it forces the encoder to be sensitive to small changes that can induce large meaning shifts. WordShuffle encodings turn out to perform significantly better on probing tasks that are related to the sensitivity property. Semantic odd man out requires identifying word insertions that are incompatible in the context, tracking word content requires knowing if a particular word is present or not, and bigram shift (BShift) requires knowing if words in adjacent positions are swapped (a subset of the training task for WordShuffle). Table 4 illustrates examples where fake sentence training is able to distinguish sentence meaning changes that results from a small change in its surface form.

**Sentence Lengths** We analyze the performance of fake sentence and Skip-thought models to understand how the models behave for sentences of different lengths. Figure 2 compares the performance of fake sentence and Skip-thought encoders binned by length. It turns out that fake sentence training performs better on longer sentences on MR but not so on the SST, even though both are sentiment tasks. The analysis for other tasks (not shown here) also indicate that fake sentence is

better for most sentence lengths but do not indicate a clear trend with respect to increasing sentence lengths.

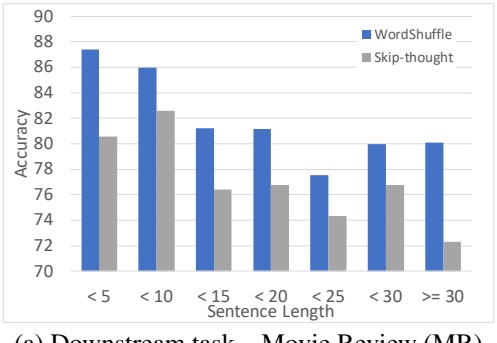 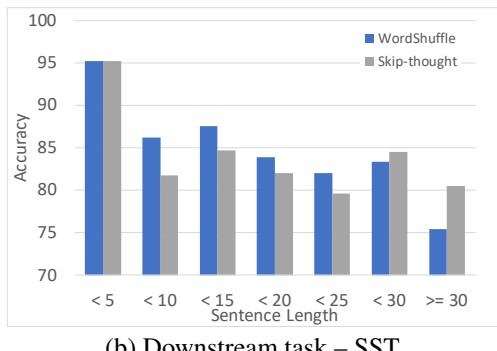

(a) Downstream task – Movie Review (MR)     (b) Downstream task – SST

Figure 2: Comparison between Skip-thought and WordShuffle encoders across different sentence lengths. Classification accuracy is shown for (a) MR and (b) SST sentiment classification tasks.

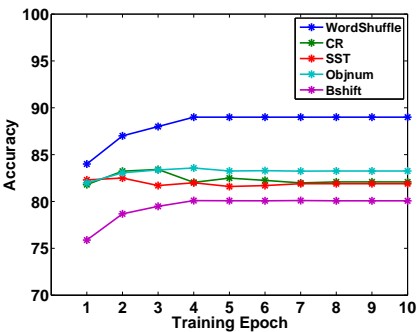

Figure 3: Figure shows the test accuracies on WordShuffle task, downstream sentiment classification task (CR, SST) and probing task (ObjNum, BShift) for different training epochs. Convergence on the fake sentence task roughly corresponds to convergence on downstream tasks.

**Relation between Fake Sentence Classification and Downstream Tasks**    In figure 3, we compare the test performance on fake sentence classification task(WordShuffle) with that on multiple downstream tasks. We notice that the encoder resulting in good performance on the fake sentence classification task results in good performance on the downstream task.

## 7    CONCLUSIONS

The effectiveness of universal sentence encoders depends both on the architecture of the encoders as well as the training task itself. Language modeling is a suitable task as it fits the generative goal of modeling the distribution of language (text data). However, tackling this in a purely generative fashion (i.e., actually generating sentences) requires large models and long training times.

Instead, we introduced a discriminative formulation of the generative task called fake sentence detection. The sentence encoders are trained to produce representations which are effective at detecting if a given sentence is an original or a fake. This leads to better performance on downstream tasks and is able to represent semantic and syntactic properties, while also reducing the amount of training needed. As future work, the discriminative setup opens up the possibility for the training to be influenced by a specific downstream target task. In particular, we can create negative samples (fake sentences) that are focused towards those specific phenomena relevant to the target task.

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
