# OpenReview forum: "Fake Sentence Detection as a Training Task for Sentence Encoding"
_ICLR.cc/2019/Conference_

### Official Review · AnonReviewer1 · 2018-10-30
**Method description confusing; empirical comparison against previous work is lacking**

**Rating:** 3
**Confidence:** 5

**Review:**

This paper proposes a method for learning sentences encoders using artificially generated (fake) sentences. While the idea is interesting, the paper has the following issues:

- There are other methods that aim at generating artificial training data, e.g.:  Z. Zhao, D. Dua, S. Singh. Generating Natural Adversarial Examples. International Conference on Learning Representations (ICLR). 2018,  but no direct comparison is made. Also InferSent  (which is cited as related work) trains sentence encoders on SNLI: https://arxiv.org/pdf/1705.02364.pdf. Again a comparison is needed as the encoders learned perform very well on a variety of tasks. Finally, the proposed idea is very similar to ULMfit (https://arxiv.org/pdf/1801.06146.pdf) which trains a language model on a lot of unlabeled data and then finetunes it discriminatively. Finally, there should be a comparison against a langauge model without any extra training in order to assess the benefits of the fake sentence classification part of the model.

- It is unclear why the fake sentence construction method proposed by either swapping words or just removing them produces sentences that are fake and/or useful to train on. Sure it is simple, but not necessarily fake. A language model would be able to discriminate between them anyway, by assigning high probability to the original ones, and low probability to the manipulated ones. Not sure we need to train a classifier on top of that.

- I found the notation in section 2 confusing. What kind of distribution is P(enc(x,theta1)|theta2, theta3)? I understand that P(x|theta) is the probability of the sentence given a model, but what is the probability of the encoding? It would also be good to see the full derivation to arrive at the expression in the beginning of page 3.

- An argument in favour of the proposed method is training speed; however, given that less data is used to train it, it should be faster indeed. In fact, if we consider the amount of time per million sentences, the previous method considered in comparison could be faster (20 hours of 1M sentences is 1280 hours for 64M sentences, more than 6 weeks). More importantly, it is unclear from the description if the same data is used in training both systems or not.

- It is unclear how one can estimate the normalization factor in equation 2; it seems that one needs to enumerate over all fake sentences, which is a rather large number due to the number of possible word swaps in the sentence,

- I am not sure the generator proposed generates realistic sentences only, "Chicago landed in John on Friday" is rather implausible. Also there is no generation method trained here, it is rule-based as far as I can tell. There is no way to tell the model trained to generate a fake sentence as far as I can tell.

- It is a bit odd to criticise other methods ofr using LSTMs with "millions of parameters" while the proposed approach also uses them. A comparison should calculate the number of parameters used in either case.

- what is the motivation for having multiple layers without non-linearity instead of a single layer?

---

### Official Review · AnonReviewer3 · 2018-11-02
**Nice simple idea but insufficient execution and discussion**

**Rating:** 3
**Confidence:** 4

**Review:**

Summary:
=======
The paper proposes a discriminative training formulation for learning sentence representations, where a classifier is required to distinguish between real and fake sentences. The sentences are encoded with a Bi-LSTM and the resulting sentence representations are then used in a number of sentence-level tasks (classification, entailment, and retrieval). The experiments show benefits on most tasks compared to Skip-Thought and FastSent baselines, and the information captured by the representations is analyzed with probing tasks, showing that they are better at capturing certain kinds of information like the presence or order of words.

The paper proposes a simple and fairly effective approach for learning sentence encoders. The basic idea is appealing and the experimental results are fairly good. However, at present it seems like more work is required for delivering a comprehensive evaluation and analysis. My main concerns with the paper are the insufficient comparison with prior work, its lack of clarity and organization in certain places, and the limited amount of work. Please see below detailed comments on these and other points, as well as suggestions for how to improve some of these issues.


Major comments:
==============
1. Better baselines and comparisons:
- The results are compared only with SKip-Thought and (the weaker) FastSent. However, there are far better models by now. First, already in the Skip-Thought paper there is a version combining Naive Bayes bi-gram features which performs much better on some benchmarks, for example that version would be better than the paper's results on MR (80.4).
- Moreover, there have been many newer papers with better results on many of the tasks [1, 2, 4, and references therein]. At the very least, mention should be made that there are better published results, and ideally there should be some comparison to the more relevant papers [1, and maybe others].

2. Paper organization and clarity:
- I found Section 2 to be unnecessarily lengthy and disorganized. It mixes motivation with modeling, introduces excessive notation, sometimes without clearly defining it (what is L_{aux}? Why is U in eq. 2 not defined on first usage?), and digresses to weakly related discussions (the link to GANs seems vague and the relation to Introspective Neural Networks is not made clear). The last paragraph is largely redundant with the introduction.
- There is also a statement that seems just wrong: "maximizing the data likelihood P(enc(x,\theta_1)|\theta_1,\thera_3)" -- the data likelihood is P(X | ...). Maximizing the encoding of x can be trivially achieved by simply having a constant encoding whose probability is 1.
- The entire Section 2 can be condensed to one or two paragraph, essentially deriving the discriminative training task in equations (1) and (2).
- On the paper organization level, this lengthy section is followed by the related work and then section 4 on "training tasks for encoders". There is again redundancy between section 4 and 2. Consider merging sections 2 and 4 into one Methodology section, where the general task is formulated, the sentence encoding (Bi-LSTM with max-pooling) and binary classifier (the MLP) are defined, and the fake sentence generation is described. This would make a better flow and remove excessive text.

3. Motivation and advantages of the approach:
- The approach is motivated by shortcomings of sentence encodings based on language modeling, such as Skip-Thought, which are computationally intensive due to the large output space and the complicated decoding process. This is an appealing motivation, although there have also been simpler methods for sentence representations that work as well as or better than Skip-Thought [1, 2].
- The second motivation is not clear to me, and the claim that "the training text collection should include many instances of sentences that have only minor lexical differences but found in completely different contexts" needs more support, either theoretical or empirical. Why wouldn't a language model be able to distinguish such differences?
- The advantages of the binary classification task make sense. The point about forcing the encoder to track both syntax and semantics is interesting. Have you tried to analyze whether this indeed happens? The probing tasks are a good way to evaluate this, but most of them are syntactic, except SOMO and perhaps CoordInv and BShift. Still, more analysis of this point would be good.
- One concern with generating fake sentences by swapping words is that it would not apply to languages with free word order. Have you considered how well your approach would work on other languages?

4. Relevant related work:
- The fake data generation resembles noise used in denoising auto-encoders. A recent application is in unsupervised neural machine translation [3], but there is relevant prior work (see references in [3]).
- The binary classification task resembles that in [1], where they train a classifier to distinguish between the representation of a correct neighbor sentences and incorrect sentences.

5. Ideas for more experiments and analysis:
- The results are fairly good by using only 1M sentences. How good would they be with the full corpus? What's the effect of training data size on the method?
- Table 4 is providing nice examples showing how the fake sentence task generates better sentences representations. Can this be measured on a larger set of examples in aggregate? Why is t-SNE needed for calculating the neighbor rank?
- Proving tasks are very interesting, but the discussion is limited. A more detailed discussion and analysis would be useful.
- Consider other techniques for generating fake sentences.


Minor comments:
==============
- Related work: the Skip-Thought decoder is a unidirectional LSTM and not a bidirectional one as mentioned, right?
- Related work: more details on supervised approaches would be useful.
- Section 4.1: how many fake exampels are generated from every real example? Have you experimented with this?
- Section 4.2 mentions 2 hidden layers in the MLP but figure 3 indicates 3 layers.
- Is there a reason to use multiple layers without a non-linearity in the MLP? This seems unusual. In terms of expressivity, this is equivalent to using one larger linear layer, although there might be some benefit in optimization.
- Table 1 seems unnecessary as there is no discussion of how dataset statistics refer to the results. It's enough to refer to previous work.
- What are some results missing in table 2, specifically SKipthought (1M) on COCO datasets?
- The paragraph on sentence encoder implementation mentions a "validation set accuracy of 89 for word shuffle". Which validation set is that? How is convergence determined for word drop?
- In analyzing sentence lengths, figure 2 shows the fake sentence to be similar to SKip-Thought on short sentences in SST. Do you have any idea why? Also, fake sentence is better than Skip-Thought on all lengths in MR, not just longer sentences, so I'm not sure there's any signal there.
- Figure 3: what is the test set for WordShuffle?
- The idea to create negative samples focused towards specific phenomena sounds like a good way to go


Writing, grammar, etc.:
======================
- Introduction, paragraph 3, last sentence: start with "The".
- Introduction, paragraph 4, first sentences: discriminative training task fake sentence detection -> discriminative training task *of* fake sentence detection
- Motivation: an useful -> a useful; we assumes -> we assume; then number -> the number; this much -> this is much
- Motivation: do not differ -> do not differ much?
- Related work: skip-gram -> skip-gram model; Training Skipthought model -> Training a Skipthought model
- Section 4: Prior work use -> Prior work uses/used
- Section 4.2: space between "Multi-layer Perceptron" and "(MLP)". This also happens with other acronyms.
- Page 6: Our models, however, train -> are trained
- Table 3 caption: is bigram in -> is bigram; is co-ordination is -> is-coordination
- Page 7: The analysis ... also indicate*s* ... but do*es* not ...
- Figure 3 caption: classification/proving task -> tasks
- References: fix capitalization in paper titles


References
==========
[1] Logeswaran and Lee, An efficient framework for learning sentence representations
[2] Khodak et al., A La Carte Embedding: Cheap but Effective Induction of Semantic Feature Vectors
[3] Artetxe et al., Unsupervised Neural Machine Translation
[4] Arora et al., A Compressed Sensing View of Unsupervised Text Embeddings, Bag-of-n-Grams, and LSTMs

---

### Official Review · AnonReviewer2 · 2018-11-03
**Simple technique, good results, not enough substance.**

**Rating:** 5
**Confidence:** 3

**Review:**

Summary: Derive sentence representations from a bidirectional LSTM encoder trained to distinguish real sentences from fake ones. Fake sentences are derived from real ones by swapping two words or dropping a single word (yielding two different models). The resulting representations are applied to various sentence classification tasks by using them as input to a logistic regression classifier trained for the task. Results are generally better than similar experiments performed with SkipThought vectors trained on the same Toronto BookCorpus.

This is a reasonable idea, and the win over SkipThought is quite convincing, but the paper is short on substance, and parts are confusing or superfluous. Some problems and questions:

1) Most of section 2 could be omitted, since it doesn’t really add insight to the well-established idea of pre-training parameters on an auxiliary task.

2) Section 3 calls the Conneau et al (2017) transfer approach supervised. It also distinguishes between semi-supervised approaches that “do task-specific adaptation using labeled data” and unsupervised approaches (including the current one) that also must do exactly that.

3) In 4.2, does the 3-layer MLP have non-linearities in its hidden layers? If so, it’s not equivalent to a single linear layer as claimed, regardless of whether a non-linearity is applied to its output. If not, there is no point in using 3 layers.

4) Section 5 gives only minimal descriptions of the tasks - often just acronym and type, presumably because they are borrowed from Conneau et al (2017, 2018). More information needs to be provided.

5) Section 6 should show the best results from the Conneau et al papers for calibration.

6) Were the baseline systems also supplied with Glove word embeddings? Do they have the same number of parameters?

7) Details of the logistic regression classifier?

8) Why train on your method on only 1M sentences, since training is fast? Wouldn’t using more text give better results?

9) Given the recent very strong results from the ELMo paper (which you cite), the current paper doesn’t seem complete without some attempt to replicate this as a baseline - eg, use a deeper encoder, combine state vectors through layers, etc. These features aren’t incompatible with your objective, which might make for an interesting extension.

---

### Meta-Review · Area_Chair1 · 2018-12-13
**Interesting idea, but missing crucial baseline**

**Confidence:** 5
**Recommendation:** Reject

**Metareview:**

This paper presents a new unsupervised training objective for sentence-to-vector encoding, and shows that it produces representations that often work slightly better than those produced by some prominent earlier work.

The reviewers have some concerns about presentation, but the main issue—which all three reviewers pointed to—was the lack of strong recent baselines. Sentence-to-vector representation learning is a fairly active field with an accepted approach to evaluation, and this paper seems to omit conspicuous promising baselines. This includes labeled-data pretraining methods which are known to work well for English (including results from the cited Conneau paper)—while these may be difficult to generalize beyond English, this paper does not attempt such a generalization. This also includes more recent unlabeled-data methods like ULMFiT or Radford et al.'s Transformer which could be easily trained on the same sources of data used here. The authors argue in the comments that these language models tend to use more parameters, but these additional parameters are only used during pretraining, so I don't find this objection compelling enough to warrant leaving out baselines of this kind. Baselines of both kinds have been known for at least a year and come with distributed models and code for close comparison.